



# Solar tracker with optical feedback and continuous rotation

John Robinson, Dan Smale, David Pollard, Hisako Shiona

National Institute of Water and Atmospheric Research, Lauder, Central Otago, New Zealand

*Correspondence to*: John Robinson (john.robinson@niwa.co.nz)

**Abstract.** Solar trackers are often used by spectrometers to measure atmospheric trace gas concentrations using direct-sun spectroscopy. The ideal solar tracker should be sufficiently accurate, highly reliable and with a longevity that exceeds the lifetime of the spectrometer which it serves. It should also be affordable, easy to use and not too complex should maintenance be required. In this paper we present a design that fulfils these requirements using some simple innovations.

Our altitude-azimuth design features a custom coaxial power transformer, enabling continuous 360° azimuth rotation. This increases reliability and avoids the need to reverse the tracker each day. In polar regions, measurements can continue uninterrupted through the summer polar night. Tracking accuracy is enhanced using a simple optical feedback technique which adjusts error offset variables while monitoring the edges of a focused solar image with just four photodiodes. Control electronics are modular, and our software is written in Python, running as a webserver on a recycled laptop with a Linux

operating system. Over a period of 11 years we have assembled four such trackers. These are in use at Lauder (45° S), New Zealand and Arrival Heights (78° S), Antarctica, achieving a history of good reliability even in polar conditions. Tracker accuracy is analysed regularly and can routinely produce a pointing accuracy of 0.02°.

## 1 Introduction

Altitude-azimuth (alt-az) solar trackers are in widespread use within the atmospheric research community. A solar tracker is

often roof-mounted, directing a vertical beam to a spectrometer in the laboratory below. Fourier Transform InfraRed spectrometers (FTIR) are commonly used to analyse the absorptions of trace gases in the slant column of atmosphere from the laboratory to the edge of the terrestrial atmosphere (Mahieu et al., 2014;Wunch et al., 2011). In a well performing system, trace gas vertical column abundances can be accurately determined and to some extent partitioned into altitude layers. These data are vital for studying the dynamics and chemistry of the atmosphere, for example to study ozone depletion

(e.g. Steinbrecht et al., 2017), green-house gases (e.g. Chevallier et al., 2011) and for the validation of similar measurements made by satellites (e.g. Dammers et al., 2017; Hedelius et al., 2019).

Inaccuracies in tracker pointing can lead to errors in calculated gas columns (Wunch et al., 2011). For example, at low solar elevations, the error in assumed absorption path length, or airmass, is significant if the tracker is pointing too high or too low

with respect to solar centre (Reichert et al., 2015) . Another type of error occurs if the tracker pointing is unstable in any direction, causing the signal intensity at the spectrometer entrance optics to vary during the observation period, resulting in





analysis inaccuracy for an FTIR measurement (Keppel-Aleks et al., 2007). FTIR measurements often run continuously throughout the day in an automated fashion. Reliability of the tracker is important, especially at remote sites without regular on-site staff present. Common failures include mechanical switches or any wires that move, for example cables from the

lower stationary portion of the tracker to the upper elevation stage, which rotates daily in azimuth. A robust mechanical design is needed to prevent loss of alignment over time. While modern control electronics have a long lifetime, finding critical replacement components in the future may be difficult. Specialised or proprietary circuit cards may be impossible to obtain after just a few years of service. Computer operating systems (OS) can update regularly, making control software potentially unsupported or proprietary drivers obsolete.


In considering the issues listed above, we present an example tracker that is sufficiently accurate and highly reliable with potential to replace or update modules in the future, ensuring a long lifetime of service.

Figure 1 shows our tracker, a standard alt-az configuration, designed to be roof-mounted and used with our MIR/NIR Bruker

125HR spectrometers (Pollard et al., 2017). The first flat mirror closest to the sun (M1), is mounted on the elevation optical rotator. M1 tracks the solar altitude (hereafter referred to as elevation). The second flat mirror, M2, is set at a fixed 45°, accepts the now horizontal beam from M1, directing it vertically downwards through the laboratory roof. Both M1, M2 and associated elevation electronics are mounted upon the larger azimuth optical rotator, which constantly tracks the horizontal (azimuth) movement of the sun. A third mirror within the laboratory and below the tracker is mounted at 45°, directing the

beam horizontally to the spectrometer input optics. Cables connect the azimuth rotator and elevation power coaxial transformer to the main electronics in the laboratory below. A laptop computer is connected to the main electronics. Figure 2 shows the completed tracker mounted on the laboratory roof. The main tracker structure is made from 10mm thick anodized aluminium plate, bolted together with stainless steel screws. The two main mirrors are elliptical aluminium-surfaced 12mm thick glass mirrors with a protective silicon oxide (SiO) coating. Our design accepts a mirror size of up to 128mm minor

axis. With minimal dimension changes to the supports it would be possible to use up to 150mm sized mirrors, fully utilising the central aperture of the tracker. The total weight of the tracker is approximately 25 kg.

The paper will now discuss a range of solar tracker design requirements and the solutions chosen when building our final design. In Sect. 2 we discuss accuracy requirements and the factors that cause mispointing in solar trackers. Section 3

describes how optical feedback is used to improve tracker accuracy. In Sect. 4 reliability is discussed, along with the use of a coaxial transformer to remove a major source of tracker failure. Section 5 describes the software and electronics in greater detail. Section 6 shows the performance of our solar tracker in terms of accuracy, reliability and longevity.



## 2 Accuracy

Spending vast resources on improving pointing accuracy is beyond the budget of many institutions. However, the tracker
should be fit for purpose, so what accuracy is required?

Our trackers play a vital role in acquiring measurement data for the Network for the Detection of Atmospheric Composition
Change (NDACC) (De Mazière et al., 2018) and Total Carbon Column Observing Network (TCCON) (Wunch et al., 2011)
databases. Measurements of solar absorption of stratospheric trace gases for the NDACC, made in the mid infrared
wavelengths (MIR) benefit from good accuracy at low solar elevations, preventing errors resulting from airmass uncertainty.
The NDACC has no specifications for site tracker accuracy. We aim to take the highest quality measurements practicable. In
the past, our main trackers used for MIR would require occasional manual adjustments to correct errors of more than about
0.1°. This level of accuracy was adequate considering the uncertainties and dynamics of the species being measured. Our
current tracker was designed to meet the site requirements set by TCCON of 0.05° accuracy, while recognizing that 0.01°
would be more ideal in achieving an airmass error of 0.1% at the lower elevation of 80 ° SZA (Gisi et al., 2011). Without
this level of pointing accuracy, it would be difficult to measure the small changes in the abundancies of the well-mixed
greenhouse gases being targeted (e.g. carbon dioxide and methane).

### 2.1 Factors affecting pointing accuracy

When no optical feedback system is used, the solar tracker is operating in dead-reckoning or passive mode. Total pointing
accuracy in this mode is an accumulation of the many sources of error of the complete design. These include, in approximate
increasing order of significance:

- Timing errors
- Algorithm errors
- Movement speed, resolution and error for the mechanical rotator stages
- Mechanical rigidity of overall structure
- Errors during mechanical initialisation (park or zero reference position)
- Levelling
- Precision of initial optical alignment

The Sun moves its full diameter (~0.5°) every 2 minutes. As a rough guide, each second of timing error is nearly 1% of the
solar diameter (or 0.0042°). Some computer clocks may drift several seconds over a 24hr period unless a timeserver or other
means of time update is used. Keeping system time accurate to 1 or 2 seconds is easily achievable, eliminating time as a
major source of tracker inaccuracy. Our design features a GPS in the top (elevation) stage of the tracker. This is polled daily





and upon startup to update the computer timeclock to the nearest second. The geographical coordinates are also updated and
the whole process is logged.

Algorithm errors are unlikely to be present in a mature design but remain a possibility. For example, refraction at low solar
elevation is significant and needs used within the pointing correction algorithm (Meeus, 1991). For astronomical ephemeris
calculations we used the Python PyEphem library (Downey, 2011;Rhodes, 2011).

Good quality rotators form the heart of any solar tracker. Gear backlash and wobble are generally much smaller than the
movement resolution of the rotator. Rotational movement speed will largely be a function of the control algorithms and
electronics and must be adequate for general tracking. If stepper motors are used, care should be taken to avoid resonance
and the resulting missed steps when moving fast. The use of acceleration/deceleration algorithms is recommended for a large
mass such as the tracker mechanics, especially when parking. An alt-az tracker can have the limitation of needing very fast
movement (in azimuth) at low latitude sites which see noontime sun directly overhead. In this case, a brief tracking error is
to be expected during the period when the rotator cannot move fast enough. Our design could allow for an alternative
solution to the overhead sun issue – simply look out the other side of the tracker by rotating the elevation mirror past noon
position. We've allowed for this "flip" position in the support hardware but chose not to implement it in software as it
required a more complex weatherproof cover. We selected Newport rotators (Newport Corporation, USA) for our design.
For azimuth we use model RV240, with a full step resolution of 0.01°. The smaller model URS75, with full step resolution
of 0.02° is used for elevation. It is usual to microstep the motor, in our case by a factor of 16, yielding a resolution of
0.000625° for azimuth and 0.00125° for elevation. Both values are more than adequate for our requirements.

Poor mechanical rigidity of the tracker structure, movement in the mirror mounts and temperature coefficients of materials
used all contribute to the passive tracking error. With conservative design these problems can be minimised. The single
sensor (or coarse flag) used on some rotators to determine park position can result in significant uncertainty of this important
initial mechanical reference. Improvements in the parking algorithm have potential to reduce this, but a better idea would be
to use a second (fine) flag on the stepper motor shaft that can be ANDed with the coarse flag. Relying on the single coarse
flag, we see a parking uncertainty in our design of approximately 0.05° (based upon observation of error offsets resulting
from consecutive parking cycles).


Precise levelling using an accurate level is an important requirement for the installation and for the initial alignment
procedure. In some installations the tracker must be mounted directly to a roof which itself is often quite unstable. A well-
designed tower, mounted to a stable laboratory floor, will produce the best results. In our example, we use a level with 1
minute of arc sensitivity (Moore & Wright, UK, model ELS). While this specification is nearly 0.02°, we discovered that in
practice it was difficult to find a suitable flat surface that produced repeatable results while at the same time attempting to
adjust the baseplate levelling screws in an outdoor environment.  Only two of our trackers are mounted to towers on solid





concrete floors. Using a precision bubble level and the somewhat coarse level-adjustment screws results in around 0.1° uncertainty in mounting the tracker truly horizontal. This is a significant source of error.

Without access to specialist tools, it can be very difficult to accurately align the mirrors in a solar tracker. Within our tracker alignment procedure, we reflect a laser-level beam through the tracker and project to a distant target. The laser spot is usually about 5mm in diameter at the target, which is at about 5m distance. It becomes difficult to resolve even 2mm of movement with this setup, which corresponds to approximately 0.02° (Tan$^{-1}$ (2/5000**))**. Subsequent use of other simple tools in the procedure, e.g. an oil bath and a precision bubble level, add to the final alignment uncertainty. In-use movement and diurnal

thermal cycling may degrade alignment further. We estimate that overall alignment accuracy better than 0.1° is difficult to achieve or maintain in our example.

From the above discussion it becomes clear the dominant source of error in a passive solar tracker is likely to be from poor levelling and sub-optimal optical alignment. A simple passive tracker will struggle to achieve even 0.1° accuracy, far from

the 0.01° required. Fortunately, good accuracy is possible if a form of optical feedback is used.

**3 Optical feedback**

Here a focused image of the sun is monitored electronically, and an algorithm makes corrections to the tracker's rotating optical stages to steer the image back to the reference position.

Traditionally, a small portion of the tracker main beam is picked off by a sample mirror prior to entering the spectrometer. This sample is focused onto a quadrant detector (a 4-element photo diode device) and the four signals are analysed. Any signal imbalance between quadrants is used to correct the tracker. The optics for this can be quite small, perhaps even mounted within the tracker itself. Signal levels from the quadrant detector are quite high because of the large surface area of the brightly illuminated detector, requiring only a small beam sample to be used. This method can be used for fast (sub-

second) control, so is useful in mobile measurement systems such as balloons, aircraft or vehicles. More recently, excellent results have been achieved using a miniature digital camera to analyse the solar image at or near the spectrometer entrance aperture (Franklin, 2015; Gisi et al., 2011; Klappenbach 2015). This method also works well for mobile systems and has the advantage of coupling the tracker optical axis directly to the spectrometer, mitigating any error caused by movement of the spectrometer relative to the tracker. The camera can be installed within the spectrometer, but only if there is adequate space

available. There is potential to use the camera to track the moon, or to perform solar limb measurements.



### 3.1 A solution using solar edge detection

We chose to use a different solution: edge detection using four silicon diodes spaced evenly around the perimeter of a large focused solar image. This method was chosen primarily because the major source of pointing errors are uncertainty in alignment and levelling. By their very nature, these errors appear as a very slow changing function, often sinusoidal, with a wavelength measured in hours. Edge detection signals are noisy compared to other methods, but the method works well if the signals are integrated over a period of many seconds or even minutes. The feedback optics were easy and inexpensive to construct and could be customised for each existing installation. Electronics and signal processing are also simple, easy to implement and comprehend. In common with most optical feedback solutions is the need to compensate for image translation through the feedback optics (i.e. mirrors and lenses), and for removing the effect of image rotation as the day progresses. The design also needs to consider the effect that clouds may have in steering the pointing away from the true solar centre.

Figure 3 shows our feedback optics. Firstly, a very small (~2mm$^2$) front-surface mirror (Fig. 3a) samples the main solar beam from the tracker above. This is aimed at a distant wall as a useful visual reference for quickly assessing tracker performance or the presence of cloud (Fig. 3b). A slightly larger non-vignetted portion of the main beam is directed into the feedback optics using a small mirror with dimensions of about 10mm x 10mm (Fig. 3c), with adjustment performed by another mirror (Fig. 3d). This sample is focused with a small spotting telescope or half a set of binoculars (Fig. 3e). The focus distance is chosen to yield a relatively large image compared to the camera or quadrant detector methods. Each of our four installations is different. Binoculars of 10 x 60 work well, as did an inexpensive 6 x 30 spotting telescope. The result should be an image diameter of between 20mm and 60mm. The larger image, in relation to the small surface area of a typical photodiode, results in better edge detection resolution. However, the larger image is less intense, requiring more electronic gain or an increase in sample pick-off area. If the image is too faint, the system would be susceptible to ambient stray light and electronic noise. The image is focused onto the diodes and amplifier printed circuit board (PCB), see Fig. 3g.

The edge detection diodes are nominally named as left, right, up and down (L, R, U and D). Focus and positioning can be checked by placing white paper in front of the diodes. The image should be centred, circular and sharp. On large images, major sunspots may be seen. The diodes are positioned to be partly illuminated by the solar disk edge. Electronic gain is adjusted to equal signal levels from the four channels, with emphasis on getting the levels in each pair equal (i.e. L = R, U = D). Physical protection and shielding from stray light are achieved using a sheet of inexpensive plastic IR filter (Optolite™ Industrial Plastics, UK) just in front of the diodes (Fig. 3g). The photodiode chosen needs an adequate surface area for the image size chosen, and a response in the near IR that matches any external filter used, e.g. PIN diode type BPW41N (Vishay).



The amplified signals are routed to the tracker electronics box and sampled at 1 to 2Hz using four channels of the 10-bit analogue to digital convertor (ADC) within the azimuth motor control PCB. The algorithm that translates these values into motor movements is described in greater detail below (Sec. 3.2). Each diode channel feeds a running average of 10 samples. This helps smooth jitter caused by atmospheric turbulence and motor stepping movement. At least one averaged signal level must meet an intensity threshold parameter. Below this level, the sky is deemed too "cloudy", reverting the tracker into
passive tracking mode. Averaged L-R and U-D pairs are analysed individually. If the signal difference within a pair exceeds a hysteresis parameter, the algorithm will act to steer the mirrors in the direction that minimises this difference. The hysteresis parameter helps prevent excessive control actions (or "hunting"). If corrective feedback action is required, the corresponding error offset variable is adjusted by 0.001° about each second. Movement priority is given to the pair with the greatest difference. If all four signals satisfy hysteresis and threshold requirements the tracker can be called "locked" and no
further corrective feedback is necessary. On a well-tuned system, with clear sky, the tracker can be locked for longer than a minute. Fast passing clouds have little effect on tracker pointing due to the long time-constant of the feedback loop. Any significant cloudiness is recorded in a logged file for post-processing of measurements and can also be used to halt automated observations until the sky is clear again. When the sky is too cloudy, the tracker reverts to passive mode, tracking the calculated position of the sun.


### 3.2 Image translation and rotation algorithm

An algorithm is needed to map the required optical feedback U-D and L-R error levels or imbalances, into the corrective movement directions for the elevation and azimuth rotators. This algorithm will account for the number and type of mirror reflections in the complete system, the position of the feedback sensors, and the number of lenses (and any prisms) in the
telescope used. For each element in the path there is at least one geometric translation required. In addition, the solar image will rotate at the feedback sensors (and the spectrometer) as the day progresses. A mathematical solution allows us to parameterise tracker optical geometry for each installation. There are two traditional approaches to finding this solution. Absolute calculation: using the knowledge of the illumination source coordinates, tracker baseplate Euler angles, and the tracker and feedback system optical components to calculate the required tracker movement, e.g. (Merlaud et al., 2012;
Reichert et al., 2015), and the method that deduces the required tracker movement empirically, by taking a small subset of deliberate mispointing measurements to calculate the required tracker angular movements e.g. (Gisi et al., 2011). We used the first approach to generate the basic algorithm, and then once-only, when the tracker is installed, manually perform deliberate mispointing to characterise the installation in terms of just two parameters that are thereafter used within the algorithm.



Figure 4 identifies the optical components and their pointing vectors. The algorithm was developed to reduce the pointing error by using the mispointing vector from the edge detection diodes. Coordinate matrix transformations **T**, due to tracker mirror reflections and rotations of the incoming radiation, are required to translate the mispointing vector ([Ex,Ey]), in the optical feedback plane, to the tracker azimuth and elevation axis angular movement reference frame ([Eaz,Eel]). [Eaz,Eel] =

**T**([Ex,Ey]). Unlike Merlaud et al. (2012), the tracker offset Euler angles are not considered in the pointing vector coordinate transformations. With adequate tracker levelling and optical alignment, the tracker baseplate Euler angles are minor and can be compensated for by active tracking. The devised algorithm is mathematically equivalent to that described in detail by Reichert et al. (2015). The readers are directed to this reference for a detailed explanation of the coordinate transformations.

The two parameters required to characterise each installation are determined as follows: Using manual buttons on webserver screen, we first attempt to move the solar image in the directions of U, D, L and R. Any rotational offset of the XY axes are ignored at this stage. A parameter string is encoded to describe in which sense a pair may needs transposing, e.g. "LR", "UD", "LRUD" or simply "N" for none. Once the movement directions map in the correct sense, the angular offset is then estimated in degrees (e.g. "60"), and this forms the rotation parameter. The rotation parameter need not be very accurate,

within 10 degrees is adequate. This is because any small trigonometric errors that accumulate are eventually corrected as if they were from a mechanical source. To reduce a mispointing signal imbalance at the diodes, the algorithm may move a combination of the azimuth and elevation rotators. This is achieved by incrementing or decrementing the error offset variable relating to each rotator. These error offsets effectively become the long-term integrator function, and mainly correspond to the systematic error in levelling and alignment. Error plots of consecutive clear days show similar behaviour (Fig 12),

opening-up the possibility of using recorded error offset values to identify alignment errors or indeed pre-correct errors during passive mode operation such as used by Merlaud, et al. (2015).

### 3.3 Initial adjustment of the optical feedback system during installation

A new solar tracker must be correctly aligned and levelled prior to initial adjustment of the feedback optics. The optical

feedback system should be disabled during this process. The solar beam from the tracker must be made vertical and the 45° degree mirror (M4 in Fig. 4) under the tracker adjusted to direct the beam horizontally to centre the solar image on the spectrometer input aperture. The tracker and spectrometer optical axes are now co-aligned. The next step is to centre the solar image on the feedback diodes using the adjustment mirror (Fig. 3d). The sky should be clear and cloud-free so that amplifier gains can now be adjusted make the four diode values equal. The threshold parameter can now be set somewhat

below the minimum expected clear-sky diode signal level. The hysteresis parameter works well if set at 10% of the threshold value. The optical axes of the tracker, spectrometer and optical feedback are now coaligned. When the feedback system is



enabled it will maintain the solar image centred on the spectrometer entrance aperture regardless of small errors in the solar tracker alignment or levelling.

## 4 Reliability

### 4.1 Factors affecting reliability

Software crashes, power failures, mechanical fatigue and exposure to extreme weather are common causes of tracker failure. Choosing a stable and mature OS is a good route to reliability. The OS and computer platform could even be of an embedded nature, removing the need for regular software updates which are an ever-present source of interruption. Over the development period, the application software and firmware should have been debugged well enough to prove reliable too.

An uninterruptable mains power supply should be used. The tracker needs protection from weather. This is achieved using good design practices such as water-proofing any electronics. The use of a custom-designed automated cover is recommended. This leaves us with the main source of tracker failure: the eventual fatigue of wiring and switches that experience daily movement. Our design eliminates this point of failure by having no moving wires or switches.

### 4.2 A solution using coaxial transformer

The upper rotating stage comprises of the elevation rotator, mirror and electronics, and needs less than 10W of power. Traditionally, this has been supplied via a flexible cable or slip rings and carbon bushes - all potential sources of failure. In 2007 we experimented with power transfer using a coaxial transformer consisting of a stationary primary winding with the secondary nested within and able to rotate freely. The primary can then be fixed to the tracker base, and the secondary is fixed to and rotates with the upper elevation stage. Figure 5 shows this prototype. Early versions used a dual primary with

outer (Fig.5a) and inner (Fig. 5b) coils, with the secondary (Fig. 5c) able to rotate within. This arrangement made the transformer more efficient, capable of transferring more power than was needed, although at the expense of reducing the available optical aperture. The final version of this transformer uses just the outer primary and has an unobstructed aperture of 150mm. Power transfer efficiency is approximately 20%. The external diameter of the transformer is 175mm, designed to fit within the chosen model of azimuth rotator. The primary former (Fig. 5d) also acts as the tracker's base. The air-cored

transformer is inefficient at the low frequencies of normal AC mains (50 – 60 Hz) but transfers adequate power at frequencies towards the limit of human hearing. 15 kHz was chosen as a compromise between efficiency, acoustic noise, potential for radio frequency emission and physical size of electronic smoothing components. The primary coil is driven with a simple oscillator feeding a power audio amplifier integrated circuit (IC) type LM3886. Power transfer efficiency is improved by resonating the secondary winding with a suitable series capacitance prior to rectification at the elevation power

supply. In the past 11 years of operation, this method of electrical power transfer has never failed. The complete tracker was tested for emissions that cause electromagnetic interference (EMI). Between 10 Hz and 75 MHz the maximum emission seen was -119 dBm/Hz. No interference was detected from the coaxial transformer.



In addition to greater reliability, the coaxial transformer has the advantage of allowing continuous 360° rotation in azimuth.
This avoids the need to reverse and reinitialize the rotator daily. In polar regions, measurements can continue uninterrupted
through the summer polar night. The elevation rotator and mirror can also rotate freely through 360° with no mechanical
obstructions. After sunset, the mirror continues to track the solar position (even below the horizon) so is ready for sunrise the
next day. Except for power-on initialisation, neither rotator need parking again. Bidirectional wireless communication is
needed between the laptop computer and the elevation stepper motor controller. We investigated transmission by optical
transducers and by modulating the power circuit through the coaxial transformer. Neither method proved easy. Instead we
used a generic Bluetooth serial link (e.g. Roving Networks, USA, model RN240F). Except for the mains power-switch, no
mechanical switches are used on the complete tracker. The ever-reliable stepper motors are used for moving the rotators.
Park/zero detection within the rotators is via contactless sensors. No electronic failures have been experienced, over 11
years, in our 4 trackers.

## 5 Software and electronics - designing for longevity

We define longevity as the ability to keep the tracker running viably for many decades to come. Factors to consider include
the likely ongoing support for the OS and application software language, communication protocols, and of course the
electronic components used. Except for specialised items, such as the rotators, the design should attempt to use generic
components and to be built in a modular fashion. The ultimate test would be attempting to build a duplicate in the distant
future. Although it would be unlikely to achieve an exact copy, with careful design, each critical module should be able to be
replaced easily with a modern version, without the necessity of redesigning the complete tracker.

### 5.1 Operating System and application software

Our tracker hardware is OS agnostic. The early version of the application was written in Visual Basic and ran on Windows.
However, the short lifespan of Windows versions, along with in-house security rules, soon caused our application software
to be obsolete. The application was rewritten in Python 2.7, with the user interface (Fig. 6) in the form of a webserver using
the Python web framework Tornado (https://www.tornadoweb.org/en/stable/). All our trackers now run on recycled laptops
running Ubuntu Linux OS. The tracker can be monitored and controlled from the laptop by browsing to the internal
webserver. The laptop can also be connected via a second network Ethernet card to the spectrometer's PC. Thus, the tracker
can be monitored and controlled from that PC too. To some extent, the tracker, Linux OS and Python application can be
thought of as a stand-alone hardware device with an embedded OS. These should never need upgrading until hardware (the
laptop) fails. When this occurs, the OS can be reinstalled, or a later version used. The tracker laptop and hardware do not
have to connect to any other device. Because no internet connection is required for operation, the complete system is largely
isolated from the outside world, thus it should prove very safe from common security threats. Our tracker hardware and



software has also run successfully on the popular Raspberry Pi platform. We have not attempted to run the current Python

application on a Windows OS. We have not attempted to run our Bruker spectrometers on a Linux OS. At present, our

preference is to keep the tracker and the spectrometer on separate PCs.

### 5.2 Main electronics

Figure 7 shows a schematic of the overall tracker and the connections between the major modules. Figure 8 shows the inside the main electronics box within the laboratory. The Ethernet to serial convertor (Fig. 8a) sends one channel (elevation) direct

to the Bluetooth module (Fig. 8b), and the other channel to the azimuth motor control PCB (Fig. 8e) with the stepper driver daughter PCB (Fig. 8d) plugged-in on top. The coaxial amplifier and oscillator, along with associated voltage regulators are included on one PCB (Fig. 8f). This PCB supplies the regulated direct current (DC) voltages needed for the Ethernet to serial convertor and motor controller. Analogue signals from the optical feedback system (diode signals) connect to ADC channels on the motor control PCB. The most likely sources of failure in the electronics bin would be the fan used to help cool the

amplifier IC and heatsink (Fig. 8g), and the mains-powered DC power supplies. In the case of fan failure, risk is mitigated using an extra-large heatsink, which in ambient lab temperatures would suffice on its own. The fan used is of high-quality unit with magnetic levitation bearings (Fig. 8h). Good quality generic DIN-standard AC to DC power supply modules are used (Fig. 8c) enabling easy replacement in the future. The total power consumption of the tracker is 55 Watts.

### 5.3 Elevation electronics

Figure 9 shows the inside the elevation electronics box. The coaxial secondary winding feeds a custom rectifier PCB (Fig. 9g). This circuit needs to cope with the voltage extremes and high frequency present in the low efficiency, high impedance transformer waveform. A switch-mode regulator (type LM2592HV, Texas Instruments, USA) handles the high voltage and keeps power usage to a minimum. The series resonance capacitor is seen at Fig.9h. The motor controller PCB (Fig. 9c) is identical to the unit used for the main electronics, good practice for keeping spare parts in common. The Bluetooth module

(Fig. 9a) is also identical to the one used in the main electronics. A GPS module (Fig. 9d), and temperature sensor (Fig. 9b), connect to spare inputs on the motor controller PCB. Figure 9 also shows how, in this version tracker, the elevation mirror adjuster micro-thread screws (Fig. 9e) are placed inside the box. This helps prevent accidental adjustment.

### 5.4 The motor controller in more detail

This is the most complex of the modules used in our tracker. The design uses the very popular PIC18F252 microcontroller IC (Microchip Technology, USA). Although our circuit design dates back to 2006, this IC is still readily available (2020). Firmware is written in C using the freely available development environment MPLAB® X IDE (Microchip, USA). The firmware code runs in a loop, awaiting a command to be received on the serial port. Commands include requests to read the GPS data, temperature sensor and other analogue and digital voltages on auxiliary pins. Another group of commands deal





with the stepper motor control. These commands range from simple step commands to more complex routines such as parking. All commands are acknowledged. The job of moving the stepper motor is performed by the plugin daughter PCB – the stepper driver (e.g. Fig. 8f). We made a separate PCB for two reasons. Firstly, this type of driver IC seemed a likely candidate to become obsolete, and secondly, this type of IC can be destroyed if abused, for example a short circuit on the cable or unplugging when in use. The IC chosen, type A3979 (Allegro Microsystems, USA), is still readily available as of

2020. Allegro make numerous similar stepper driver ICs and hobby-electronics suppliers use these (and other manufacturers' drivers) in easy-to-use modules for robotics and 3D printers etc. There is little doubt that suitable stepper motors and drivers are going to be available for decades to come.

**5.5 Data communication methods and protocols**

Serial communication to motor controllers is via RS232. This protocol retains strong support throughout the instrumentation

industry. RS232 enables easy low-level testing and development of the tracker without the additional complexity of, for example, USB. RS232 ports are still present on most new desktops and additional plug-in cards are readily available. Bluetooth modules are used to link data to the elevation stage. Bluetooth appears to be supported well into the future as it continues to be present in modern consumer devices. Our data speed is low (9600 baud). Many other forms of low-power radio link could be easily be used instead. In its more basic form, our tracker will run directly from two RS232 ports on a

desktop PC and not need to use an Ethernet connection. However, using an Ethernet to serial convertor is a good solution if a laptop (which generally lack RS232 ports) is to be used. In our installations, an Ethernet to serial converter, laptop and the spectrometer all share the same subnet and connect to the spectrometer's Windows PC via a second Ethernet card. This makes for a very tidy and useable system. Ethernet continues to be well supported. In the future it might be necessary to upgrade to a different model Ethernet to serial convertor, but due to the modular design philosophy, this is a small task.

**6 Results**

Over a period of 11 years we have assembled four solar trackers. These are in use at Lauder (45° S), New Zealand and Arrival Heights (78° S), Antarctica, achieving a history of good reliability even in polar conditions. Tracker accuracy is analysed regularly on two of our trackers and can routinely produce a pointing accuracy of 0.02° from solar centre.

**6.1 Accuracy achieved**

We have two methods for monitoring accuracy of our trackers. The first is simply by observation. For this we project a sample of the tracker's vertical beam with a very small (~2mm$^2$) chip of front-surface mirror (Fig3a). Aimed at a distant wall, this acts like a pin-hole camera producing a reasonably well-defined solar image if projected onto a white piece of paper. The circular image can be outlined in ink and any deviations can be assessed in terms of relative movement of the





solar diameter. An error of more than 5% (0.025°) is very apparent, yet in practice never seen. In a similar manner, the focused solar image at the FTIR entrance aperture can be viewed if the instrument source compartment has a transparent cover. Our four FTIRs have these covers.

The second method for assessing accuracy involves post-processing of our routine TCCON FTIR measurements to assess
solar-telluric wavelength shift (S-G shift). S-G shift is obtained from analysis of solar absorption features (so-called Fraunhofer lines) in the oxygen column retrieval in the 7882 cm$^{-1}$ band (Wunch et al., 2011). This analysis uses Doppler wavelength shift of sunlight caused by the relative rotational velocities of the Sun and Earth. Absorptions within the solar atmosphere show greatest shift along the equatorial edges of the solar image where the apparent velocity is highest. The main limitation with this method is that sensitivity exists only in the direction normal to the solar polar axis (Reichert et al.,
2015). However, the cause of any tracker error is unlikely to fall exactly (and only) along the polar axis for long periods and the solar image also rotates as the day progresses. This analysis remains very useful. The method does require the use of a spectrometer capable of acquiring suitable high-resolution solar spectra and the user must have the skills to perform the analysis involved. The results are not in real-time. Correct analysis relies on the solar image beginning centred on the spectrometer aperture. If this is not done, then the analysis may show a high pointing error even when the tracker is
performing very well. We perform S-G shift analysis as part of routine TCCON data processing soon after the day's data are uploaded to our server, and the results are displayed as a web-plot for easy viewing.

Figure 10 displays S-G shift on a clear-sky day during an intercomparison between our two Bruker IFS125HR spectrometers (with solar tracker names "Tracker1" and "Tracker2"). With perfect sky, clean mirrors and both spectrometers well-aligned,
this plot shows tracker pointing accuracy (for both trackers) better than 0.02°. A lone red dot represents the first measurement performed at sunrise, before the Tracker1 feedback was locked. The TCCON pointing accuracy requirement of 0.05° is easily met.

This sensitivity of using S-G shift as a pointing diagnostic can be tested by intentionally miss-pointing the tracker by a small
amount. Figure 11 shows the results of shifting the solar image by (approximately) 10% of a diameter (0.05°) and 5% (0.025°). The exact orientation of the solar polar axis was not known, instead miss-pointing was performed in two orthogonal axes so that a range of positive and negative spectral shift errors would be captured. Figure 11 shows that the sensitivity of this method is adequate to detect such errors. It is also useful to see the magnitude of error correction performed by the optical feedback system. The offset error variables represent the corrections the feedback system needs to
add (or subtract) from the calculated ephemeris for positioning the azimuth and elevation rotators in order to keep the solar image centred within the 4 photodiodes. The offset errors slowly change over the day and are logged to file. They provide a good indication of the state of mechanical alignment and/or levelling error. Figure 12 plots these offsets for 2 different trackers during a spectrometer intercomparison. The test was done again 6 months later. Figure 12a plots these offsets for



Tracker1. Some change is seen over the period, although part of the shift between date pairs is caused by the operator
performing an offset-save within the application. This action adds any instantaneous offsets to the setup offset parameters,
and re-zeros the instantaneous offsets. The overall pattern, especially evident in the azimuth (AZ) offset, shows the slow-
changing nature of the error sources involved. Figure 12b shows a similar plot for Tracker2. The relatively small offsets
indicate good alignment and/or levelling of this tracker. Because both trackers indicated a similar (and good) level of
accuracy at the same time (e.g. Fig. 10), we can infer that Tracker1 is not well aligned or is no longer level. This reinforces
how effective our simple optical feedback system can be. However, it is worth mentioning that if the errors get much larger
than the solar diameter (0.5°) then this feedback system may not automatically achieve lock (because the solar image could
miss all diodes). Tracker1 is close to this situation and probably needs re-levelling or alignment soon.

Solar tracker long term accuracy and stability assessment was performed by identifying periods of clear sky (using all-sky
camera images) and plotting the analysed S-G shift daily averages from measurements made on these days. Since FTIR
measurements are taken automatically, they may still include some observations affected by cloud (or other weather
conditions such as haze or wind). However, our current TCCON spectra acquisition and processing procedures have further
quality assessment/quality control (QAQC) measures to eliminate the majority of weather affected observations.

S-G shift analysis is also affected by events not related to the tracker. For example, any adjustments (planned or accidental)
to the spectrometer perturbs the instrument. Because the spectrometer is supported by numerous coil springs, even simply
leaning on the instrument, or bumping it, results in moving the optical axis away from its previous location. The effect of this
makes the solar image no longer centred on the entrance aperture. The S-G shift analysis captures this as a tracker pointing
error. In such cases, the solar image can be manually re-steered back onto the centre of the entrance aperture using the 45-
degree mirror underneath the tracker tower. A laboratory logbook records some of these events. Figure 13 displays the S-G
shift, over a 24-month period when Tracker1 was in use. Logbook events often correlate to a step in S-G shift, especially
when the 45-degree mirror underneath the tracker is adjusted. Days with high standard deviation are the result of cloud either
affecting a measurement directly or delaying the initial feedback lock of the tracker (and thus are a true pointing error). High
standard deviation occurring on 1st May 2020 is the result of intentional miss-pointing experiments.


## 6.2 Reliability achieved

We've experienced few interruptions due to failure of our solar trackers. Typically, the trackers run continuously for many
months with no operator intervention. The only failure seen appears as a stalled application. To date we've been unable to
locate a common cause for this, but suspect it arises from code used in our application or data loss in either the Ethernet or
Bluetooth links. To put this in context, the spectrometer and PC require restarting much more often than our tracker. The



Newport rotators are of excellent quality and are running at such low duty (1 turn per day) that we do not expect them to wear out for a long while.

### 6.3 Longevity achieved?

Development started in 2006 with the first tracker in constant use since 2009. Our most recent unit was built 2016-2017. The basic design has remained constant, with the only significant change being recoding of the software from Visual Basic to Python. A range of spare parts has been kept, for example: a complete spare electronics bin has been made and is kept at our remote Antarctic laboratory, along with a spare laptop. The electronics bin is completely interchangeable with any other tracker. The motor controller PCB is compatible with azimuth and elevation electronics, and a range of other PCBs are kept

as spares. A spare pair of Bluetooth links is also kept on site at Lauder.

### 6.4 Other benefits

Raw feedback diode values are logged approximately each second. These data files are downloaded and used in the spectra QA/QC process to automatically cull measurements made during periods of cloudiness. In addition, the automated

scheduling software used to take FTIR measurements (Geddes et al., 2018) can inspect this file in real time and will only initiate observations when the sky is not flagged as "cloudy". Using these methods, we've increased our measurement density and quality while at the same time reducing the manual effort required in pre-processing quality control. The webserver user interface has proven easy to use. When networked, the tracker is easily controlled from a remote location. The Python application and electronics can be adapted to drive other types of alt-az trackers, for example we successfully

reused an existing tracker dating from the 1980s. A configuration file allows for alternative resolution rotators to be used.

### 7 Conclusion and discussion

The design of this solar tracker has proven successful for its intended use. We conclude that:

- The use of a coaxial transformer to transfer power to the upper (moving) stage has proven to be very reliable, with the added advantage of 360° continuous rotation.
- A form of optical feedback is needed to meet today's tracker accuracy requirements.
- The simple edge-detection method of optical feedback works very well.
- A basic webserver makes a versatile user interface for a solar tracker.
- Analysing the solar spectra for S-G wavelength shift is a good method of assessing tracker pointing accuracy.



- The Newport optical rotators used in this design have proven to be long-lasting and reliable, with adequate mechanical resolution when motors are micro-stepped.

Due to the modular design philosophy, the use of basic communication protocols and commonly used software and OS achieves a long-lasting and easily maintainable system. The design detailed in this paper was made for FTIR solar spectral measurements for NDACC/TCCON purposes, but in principle could be easily adapted for other uses. Mirrors, rotators and even the coaxial transformer could be scaled to suit larger or smaller applications.

Our current tracker accuracy is limited by the stability and optical resolution of the feedback system. Although the current optics meet our accuracy requirements, they are far from perfect. Further experimentation using improved mechanical stability and better optics could well be beneficial, especially for when the tracker must be operated in dead-reckoning mode (e.g. using moonlight as a source). Tracker levelling accuracy could be improved using fine-pitched adjustment screws on the tracker tower baseplate. The use of a permanently attached digital accelerometer may enable easier levelling and ongoing monitoring. Mirror stepping movements can be perceived in the reference solar image projected to the distant wall, although at times it is no worse than atmospheric turbulence. This movement is the result of the approximate 1-second loop cycle, meaning multiple steps are often needed to track the sun, resulting in a larger (accumulated) single movement than if done several times per second. The present generation of motor controller has rather verbose commands and is quite slow in stepping. The Python application and the firmware of the motor controller have good potential for improvement in this area, especially if greater movement resolution was thought necessary.

Because of the long integration time and "nudge and wait" approach of our feedback method, it takes 1 or 2 minutes to achieve initial "lock" after the first clear sun appears. This could be sped-up by saving and reusing offsets from a recent clear day. In practice, few observations are lost with this deficiency because the sky can remain unsuitable for optimum measurements for long periods, and as the sky gradually clears, the tracker is continuously decreasing the pointing error. Once the sun is finally clear, lock is quickly achieved. The algorithm that averages the diode values and uses the threshold and hysteresis parameters could be improved to provide better accuracy. For example, signal strength history could be used to auto-set the threshold and hysteresis parameters. This would consider the reduction in signal level under uniform hazy sky, or as mirrors slowly accumulate dust between routine cleaning.

There are clearly potential benefits for switching to a camera-based feedback system, for example it should be easier to perform solar limb measurements (intentionally pointing off solar centre to separate solar and terrestrial absorptions). The tracker has been successfully used for measurements using moonlight, but this was done in passive (non-feedback) mode, requiring careful setup and manual adjustments every hour to account for inherent misalignment/levelling of the tracker. Lunar measurements with optical feedback would be much easier using the extra gain of a camera-based system. If a camera



was used, we could retain the philosophy of correcting for a slow changing error function ("nudging" the error offset variables). A camera system, if imaging the spectrometer aperture, provides the important direct coupling of the tracker and spectrometer optical axes that our current system lacks. We do intend to move the optical feedback system to mount directly to the spectrometer, effectively tying the optical axes of the tracker and spectrometer together. This should reduce the steps in accuracy seen in Figure 13. Our main code is written in Python 2. A move to Python 3 would be another worthwhile investment.

It would now be practical to produce parts of the solar tracker using modern 3D printing processes and new materials such as glass or carbon fibre-reinforced nylons. The mechanical structure including coaxial transformer components and the elevation electronics would print easily, enabling the design to be easily replicated or scaled to suit other applications. For lower resolution applications that are cost-conscious, the elevation rotator might be replaced with a direct-drive stepper motor using a higher microstepping value (256 is now possible).

With the tracker design working well, the next step was to design a matching automated cover (Fig. 14). This needed the ability to rotate continuously in azimuth and to protect the tracker from extreme weather. The cover was completed in 2013 and its use has resulted in a significant increase in our number of measurements.

**8 Author contributions.**

JR designed and built the solar tracker along with the electronic PCBs and optical feedback system. JR programmed the embedded devices and computer software. DS developed the active feedback algorithm. DP provided S-G shift spectral analysis. Plots of S-G shift were produced by HS.  JR wrote the paper with contributions from DS.9 Competing interests

The authors declare that they have no conflict of interest.

**10 Acknowledgements**

We wish to acknowledge colleagues from NIWA who have contributed to the successful outcome for our solar tracker. Alan Thomas and Paul Johnston contributed their extensive knowledge of optics to the initial design of our prototype. Brian Connor identified the need for a more accurate solar tracker and gained the seed funding and support needed to start the project. Alex Geddes and Bruno Kinoshita contributed important Python coding skills to the final application. Brendon Smith and colleagues at NIWA Instrument Systems assisted with serial manufacture (CAD/CAM) for some hardware parts.

We also thank Antarctica New Zealand for providing support for the FTIR measurements at Arrival Heights, which included testing tracker performance in polar conditions. Tracker development was core-funded by NIWA through New Zealand's Ministry of Business, Innovation and Employment Strategic Science Investment Fund



We were also well supported by our TCCON research colleagues at CalTech, USA. Dougal Hiscock (Engen Engineering, New Zealand) greatly assisted with design and manufacture of our custom mirror mounts.



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




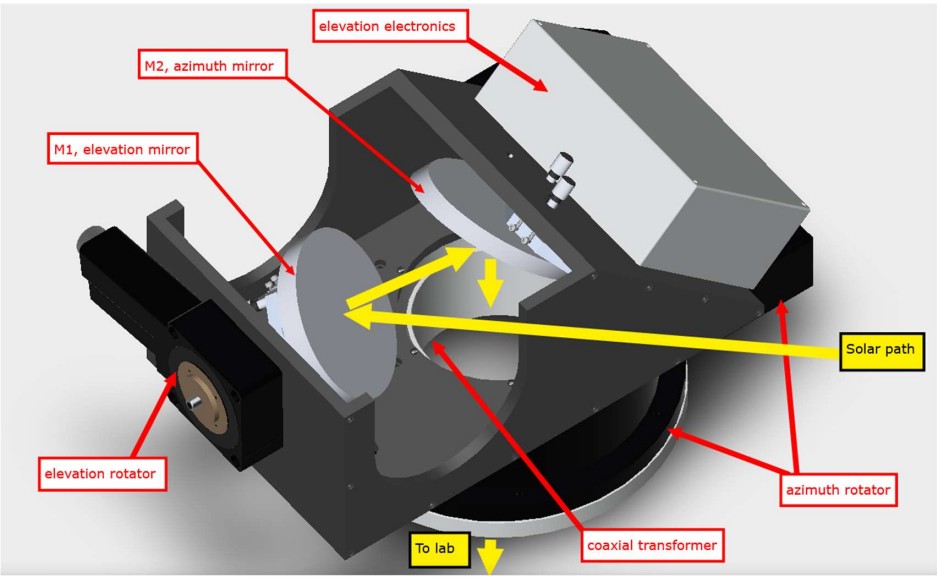

Figure 1: Our alt-az solar tracker CAD model, with major components identified.





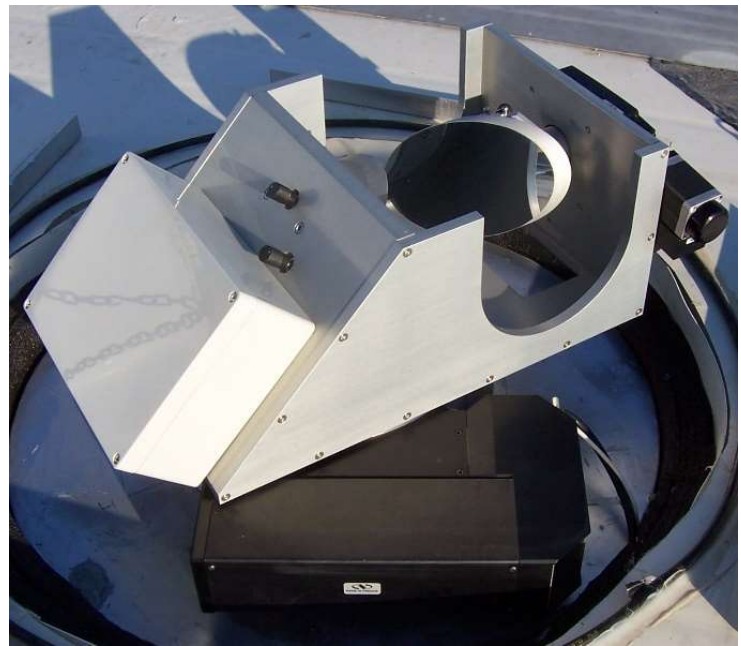

**Figure 2: The completed tracker (Lauder, New Zealand) installed at roof level, on a tower mounted to the laboratory**

600                                                                        **concrete floor.**





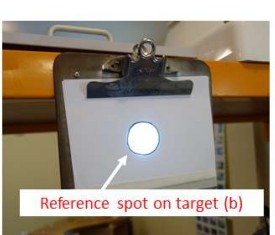
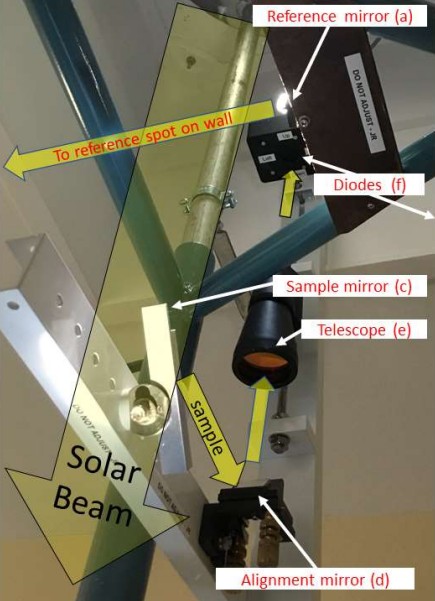
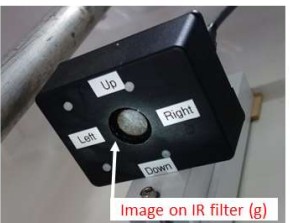

**Figure 3: The optical feedback components identified. The main view is from the laboratory and looks upwards towards the roof-mounted solar tracker.**




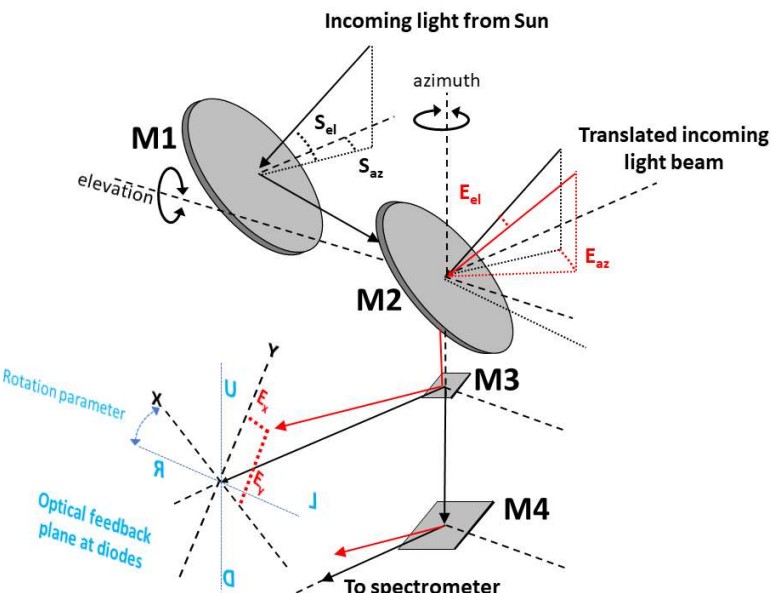

**Figure 4: Geometry of the Lauder solar tracker optics for the simplest configuration. Focus optics are not shown. M1 and M2 are the tracker alt-az mirrors. M3 is a small flat mirror to direct a portion of the incoming beam to the optical feedback system. M4 is a large flat mirror directing the solar beam into the spectrometer. [Sel,Saz] is the calculated (solar pointing) incoming beam vector. X-Y is the planar coordinate system of the optical feedback system, shown here prior to correction with the installation-specific rotation parameter. [Ex,Ey] is the mispointing vector. [Eel,Eaz] is the mispointing vector transformed into the tracker vector reference frame.**



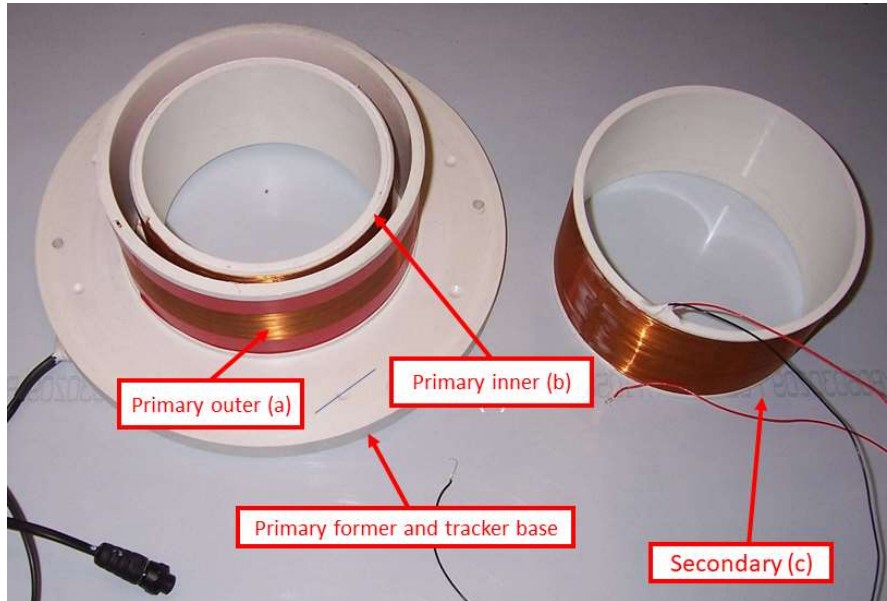

**Figure 5: Coaxial transformer parts**



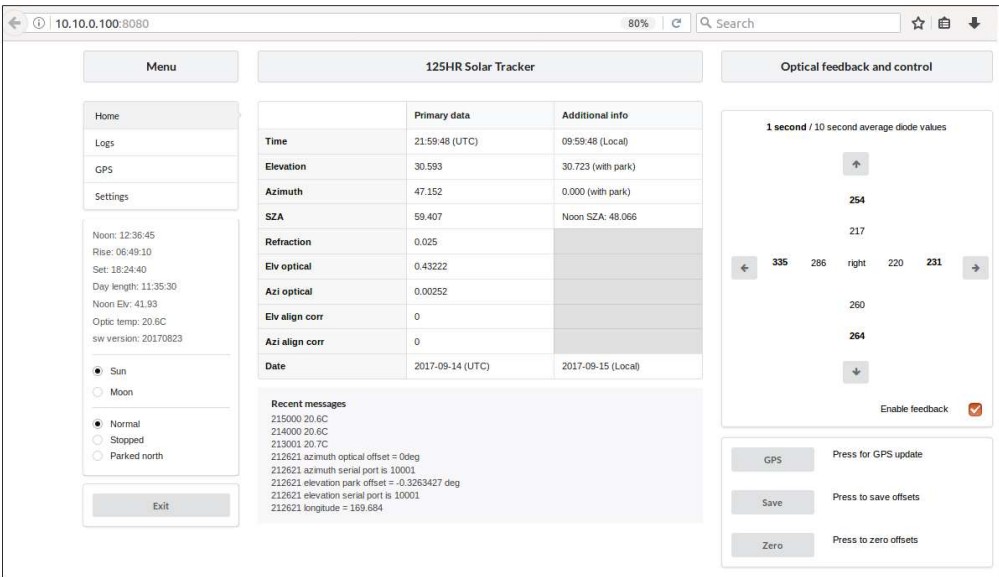


**Figure 6: The Webserver user interface**



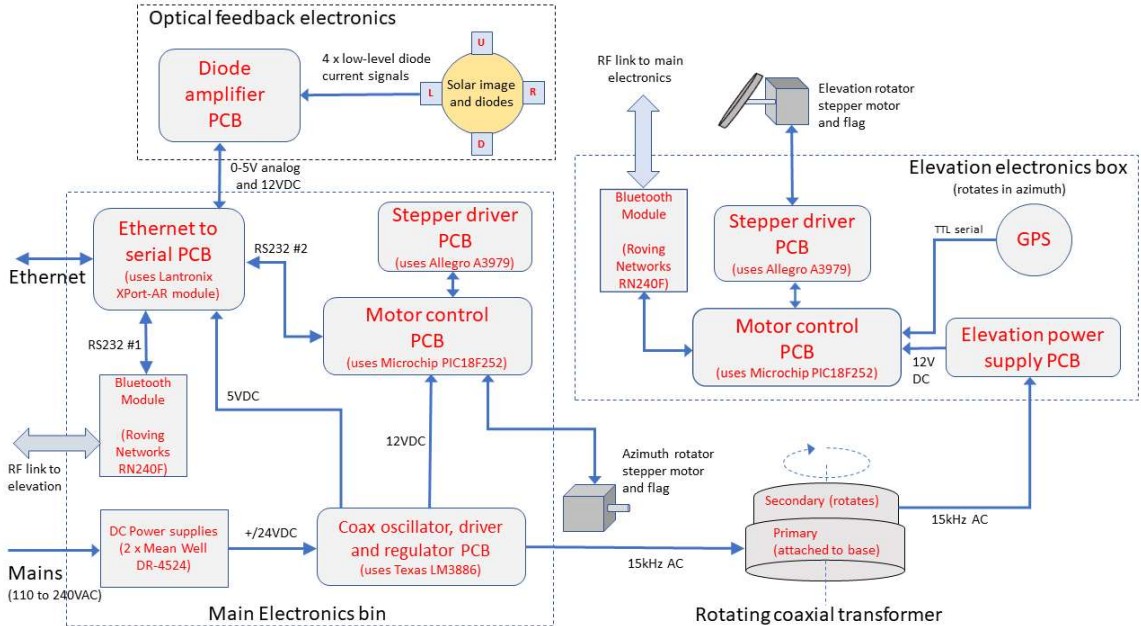

**Figure 7: Schematic of the electronic parts and connections**






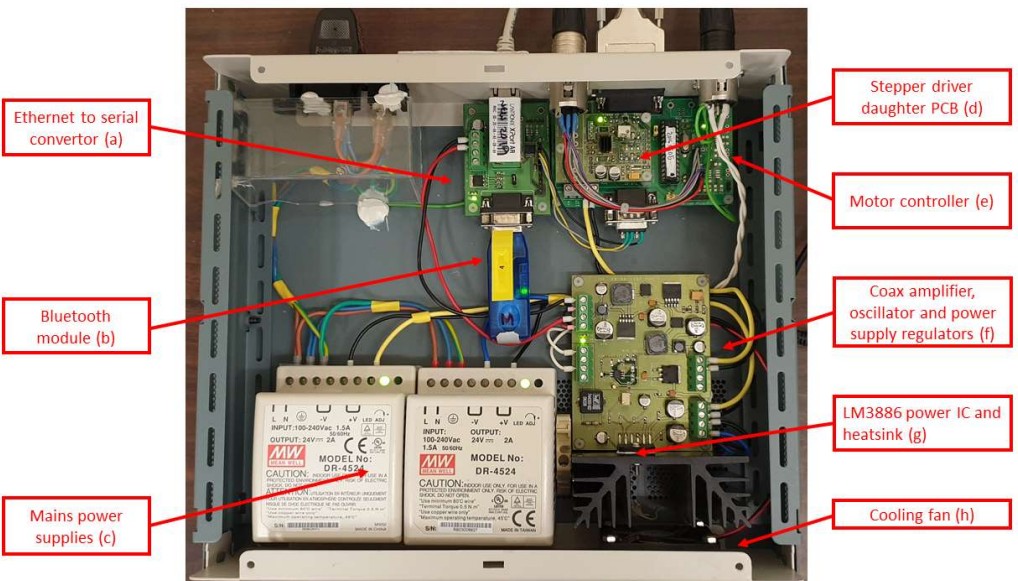

**Figure 8: Main electronics bin.**





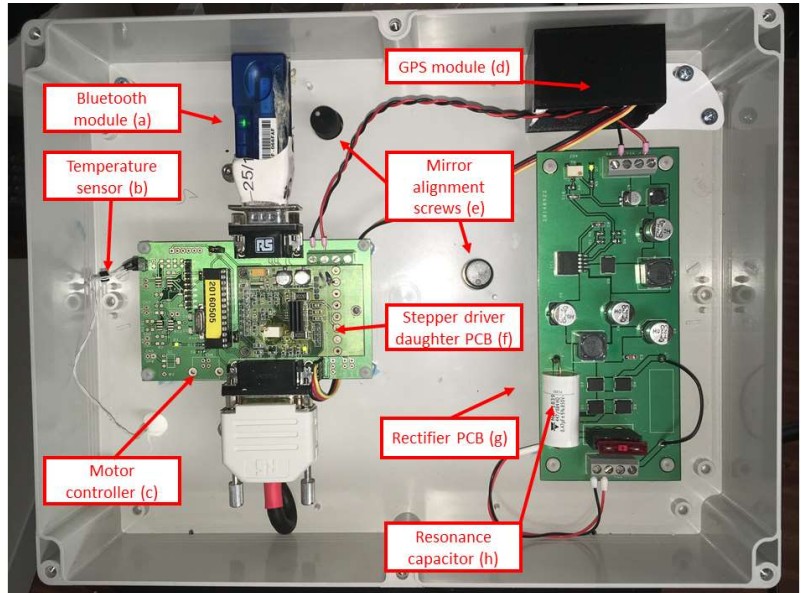


**Figure 9: Elevation electronics**

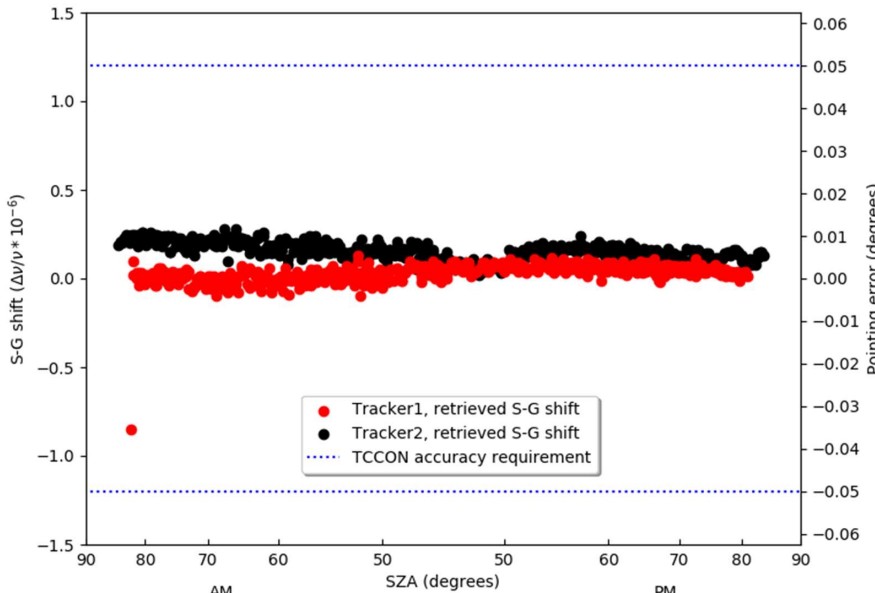

**Figure 10: Indicative accuracy of two solar trackers, inferred using S-G shift analysis under clear sky conditions during spectrometer intercomparison, 29th March 2019. "Tracker1" and "Tracker2" are local tracker identifier names. The blue dotted line indicates TCCON accuracy specification.**





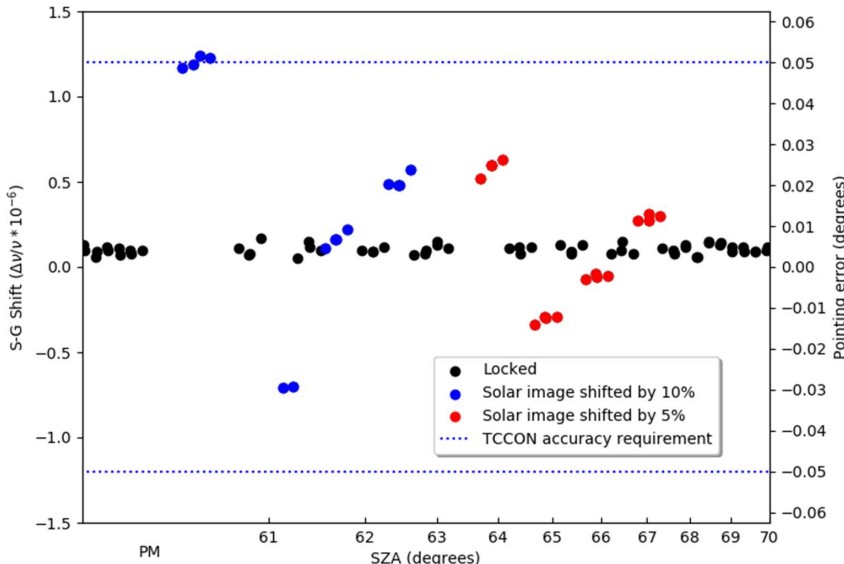

**Figure 11: Intentional miss-pointing tests in four quadrants (L, R, U, D) by 10% (blue) and 5% (red) of the solar image diameter. Tests performed 1st May 2020 using Lauder Tracker1.**




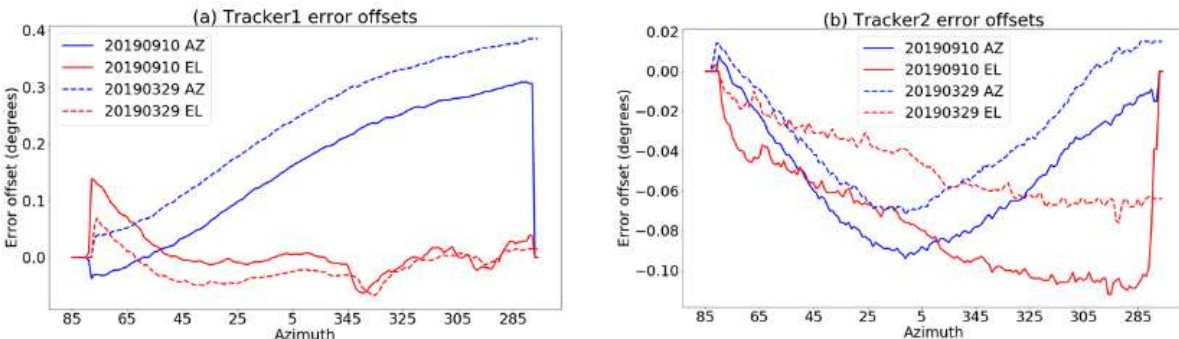

**Figure 12: Error offsets of Tracker1 (a) and Tracker2 (b) 6 months apart. Blue lines plot azimuth (AZ) offsets and red lines show elevation (EL) offset**




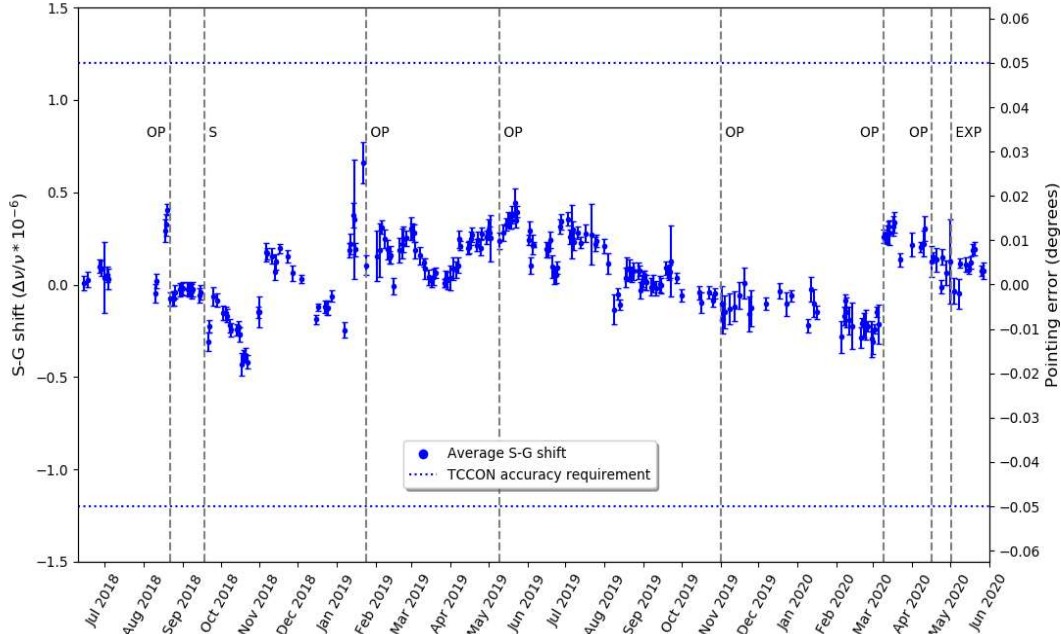

**Figure 13: Retrieved S-G shift for measurement days over a 24-month period using Tracker1. Each point is an average value for TCCON spectral measurements taken during periods that were detected as clear sky. Vertical lines mark dates when specific**
**maintenance events were performed on the equipment. Cleaning or adjusting the 45-degree mirror optics in front of the spectrometer was performed on events marked as "OP" and a major alignment performed on the spectrometer ("S"). Days with high standard deviation are generally the result of cloud-affected measurements, and this is confirmed when looking at the day in detail. On 1st May 2020 ("EXP") we performed the intentional mispointing experiments, and this day also shows high standard deviation, which was to be expected.**






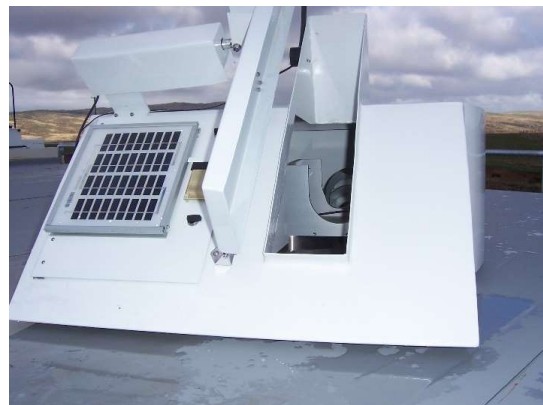

**Figure 14: Our completed tracker inside matching automated weatherproof cover**