# Peer review of "Solar tracker with optical feedback and continuous rotation"

_Atmospheric Measurement Techniques, 2020_

## Referee Comment (RC1) · Anonymous Referee #1 · 21 Jul 2020

The manuscript under consideration presents a carefully designed and tested solar tracker for stationary ground-based direct sunlight observations. Such solar trackers are essential for greenhouse gas observing networks like the Total Carbon Column Observing Network (TCCON). The pointing accuracy of solar tracker must be better than 0.05° to reach the measurement precision such networks strive for. To fulfill this obligation, the authors developed a solar tracker with an optical feedback loop based on solar image edge detection, achieving a pointing accuracy of 0.02°. Furthermore, their design is mechanically robust and and designed for long-time use, avoiding failure due to mechanical fatigue of moving wires by replacing them with a coaxial power transformer. This solution also enables the continuous rotation around the azimuth axis, which is especially useful in polar regions. Alongside the carefully chosen hardware which is very likely to be supplied in the future, these developments provide an excellent solution of a durable, reliable, and precise solar tracker.

The topic addressed in this manuscript match well with the scope of AMT. The text sometimes becomes hard to follow, mostly because a lack of clear structure. I recommend ordering the information more hierarchically to avoid raising questions to the reader which only get answered much later in the text, especially in section 3 (Optical feedback). Also, many adjectives used either need to be more precise or put in context, otherwise they provide no distinct information (e.g. good, large, small).

**General comments**

Line 115 ff: Re-structure paragraph. First, give the numerical value of the error when declaring it as significant, then describe how you arrived at this value and present your solution strategy last.

Line 130 ff: Please describe your alignment process in more detail. What would be specialist tools, or is it the level laser you've used? Why is the movement of 2 mm of a 5 mm diameter spot difficult to resolve?

Line 157 ff: The following chapters 3.1, 3.2, and 3.3 are hard to comprehend since the information are badly structured. I'd very much appreciate an explaining sentence on how your feedback system works in the first paragraph. If I've understood it correctly, you are not actively tracking the Sun solely by the optical feedback system but are primarily relying on astronomical calculus. The optical feedback than adjusts the error offset vector in a manner that the astronomical calculations provide a sufficiently precise pointing, which is the state you call "locked". Please introduce your concept in the beginning before going into details.

Line 177: How do you approach this trade-off in image size and intensity?

Line 183: The partly illumination of the photodiodes is important to get the basic idea of the feedback loop, mention it earlier.

Line 190 ff: Algorithm description hard to comprehend. Please introduce in a clear way
- what's the threshold and hysteresis parameter?
- how does the algorithm steer the motors and what happens if parameter bounds are crossed?

Line 218: Name the parameters you use for the feedback correction and explain why they are sufficient.

Line 230: Again, you use "the parameters" without naming or explaining them. This would help a lot following your characterization approach.

Line 318 ff: Give manufacturer and description to each compartment, e.g. Bluetooth module

Line 393: On which stations are these Trackers?

Line 404: Description of parameters must be given in algorithm chapter.

Figure 13: How long are the averaging periods?

**Specific comments**

Line 12/13: Use of "simple" and "just" unnecessary.

Line 48: Can you provide the reflectivity range of the mirrors?

Line 59: Chapter 2 only covers mispointing of passive solar trackers, please add this.

Line 67: Define "vital" role, sth. like "Our trackers enable direct sun observations at sites XY..."

Line 76: Define "small changes"

Line 78: Add "passive" to the chapter title

Line 97: "required" instead of "needs used"

Line 139/40: remove "simple" and replace "good" with something more descriptive

Line 158: Either remove "large" or put it into context

Line 169: Remove "very"

Line 173: "[...] adjustments performed by another mirror." The figure caption calls this the alignment mirror, since it, as explained later, is used to adjust the optical axis onto
the feedback plane. Be consistent with naming and explain parts at first occurrence in the text.

Line 186: What's an adequate diode size relative to image?

Line 213: Sentence structure is weird, either introduce both approaches using colons or neither one, I'd go with the latter.

Line 231 ff: Mention that you move the image on the feedback plane. Also, describe in further detail how you determine the angular offset.

Line 239: "consecutive" implies successive days, but 6 month lay in between, I'd recommend another adjective.

Line 259: Can you give a notion how reliable the software has proven to be, e.g. a number of failures during the 11 year period?

Line 261: Maybe give examples, see e.g. Heinle and Chen (2018) (doi: 10.5194/amt-11-2173-2018)

Line 277: remove "simple"

Line 284: make clear what you're comparing to with "greater".

Line 327: Define why the modules are "Good"

Line 390: remove "very"

Line 444: Your title poses a question, give a decisive answer in the following paragraph.

Line 467: "works very well"  $\rightarrow$  e.g. "surpasses precision requirements"

Line 469: replace "good"

**Technical corrections**

Line 55/56: space between numbers and units "2 mm", please add the space everywhere in the text. Line 128: horizon\*t\*al Line 287: so \*IT\* is ready Line 291: present tense "use" Line 326: "of" should be an "a" Line 330: "[...] inside \*OF\* the [...]"
Line 333: space after "Fig." Line 372: space after "Fig." Line 575/576: DOIs in references missing Line 580: "&ndash" in reference

---

## Referee Comment (RC2) · Anonymous Referee #2 · 11 Aug 2020

While it appears that solar tracker design continues to be a game of reinventing the wheel, the authors of this manuscript describe a handful of useful innovations not typically employed in other trackers. Notably, the use of the coaxial power transformer – allowing 360 deg rotation without the typical wear on moving wires – is a significant improvement on past designs. The system is shown to be precise and reliable enough to be used in remote locations with little need for operator input following the initial alignment.

However, I share the concerns of Reviewer #1 that certain sections could be significantly improved by re-writing with a better focus on the narrative progression. Also, while the authors dive in with significant details on some portions of the design, other sections read as if their inclusion was simply an afterthought (e.g., two brief sentences

regarding the cover for the tracker – a critical component of any remotely operated system.)

The manuscript will certainly be very worthy of publication upon addressing the issues laid out by Reviewer #1 (whose comments I fully agree with) as well as these additional specific comments:

Lines 64-77: Section 2 Intro – I suggest improving the clarity of this section intro with a more direct connection between the question posed on line 65 (accuracy required) and the answer of 0.05 deg given on line 74.

Line 70: comma after MIR

Line 98: "...needs used within..."

Line 99: please include version of PyEphem

Line 163: "...are also simple, AND easy to implement and comprehend"

Line 201: How does the system respond to a partially obstructed solar disk (sliding along the edge of a cloud over many seconds). And how does it recover after such an episode? Could you increase the 0.001deg/sec offset adjustment based on the degree of signal difference within a sensor pair to more quickly recover?

Line 230: Reference to webserver before the webserver is introduced.

Line 250: The threshold level setting procedure / choice of hysteresis parameter is not particularly clear. How do you define "works well" when set at 10%?

Line 260: drop s from mains?

Line 288: Does simply passing the reference point in normal operating mode sufficiently maintain the rotator stage's knowledge of its position? While such knowledge may not be necessary in active tracking, is it not still needed in the passive ephemeris tracking mode?

Line 333: Space after Fig.

Line 367: Why is the tracking accuracy not regularly analysed for the other two trackers you have built?

Line 370: While I applaud the addition of the wall image in the design – certainly a very useful visual for a lab operator in the room – I would hardly refer to it as a valid means of monitoring the tracking accuracy, especially for a design which you emphasize can be efficiently operated remotely. I recommend de-emphasizing the quick visual method, and perhaps more clearly highlighting the consistency of the offset errors as a function of alt/az.

Line 410: Given that the offset errors are a function of alt/az, how useful is a single alt/az setup offset parameter pair? I imagine these much be updated throughout the year?

Line 516: This manuscript regularly emphasizes the remote operation of this tracker. Thus, a reliable automated cover is a critical component of the design. I believe it would be very appropriate and desirable to include more information on the cover design in this manuscript.

Fig 13: Despite being well within the required accuracy requirements, I would like the authors to speak to the changes in S-G parameter seen in November 2018 and in early Jan 2019 ahead of the optics cleaning

---

## Author Comment (AC1) · 3 Sep 2020

Authors response to Anonymous Referee #1

We thank the reviewer for taking the time to read, comprehend and provide useful comments regarding our manuscript. In this document, the reviewer's comments are in italic, and the authors' responses follow each comment in plain bold text.

***General comments***

*The text sometimes becomes hard to follow, mostly because a lack of clear structure. I recommend ordering the information more hierarchically to avoid raising questions to the reader which only get answered much later in the text, especially in section 3 (Optical feedback). Also, many adjectives used either need to be more precise or put in context, otherwise they provide no distinct information (e.g. good, large, small).*
**Response: This observation was also noted by Referee #2. Section 3 (optical feedback) has been rewritten in light of these comments. We have also removed or better defined the majority of the adjectives listed. We have appended the revised draft for this section at the end of this document.**

*Line 115 ff: Re-structure paragraph. First, give the numerical value of the error when declaring it as significant, then describe how you arrived at this value and present your solution strategy last.*
**Response: We have restructured the paragraph which discusses the parking error. It now reads:**

**"The single sensor (or coarse flag) used on some rotators to determine mechanical park position can result in significant uncertainty of this important initial reference. For example, we see a parking uncertainty in our design of approximately 0.05° (based upon observation of error offsets resulting from multiple consecutive parking cycles). It may be possible to reduce this uncertainty using a second (fine) flag on the stepper motor shaft that is then ANDed with the coarse flag, resulting in a finer resolution for this position. We have not implemented this solution at present, although we allow for this option in our electronics design."**

*Line 130 ff: Please describe your alignment process in more detail. What would be specialist tools, or is it the level laser you've used? Why is the movement of 2mm of a 5mm diameter spot difficult to resolve?*
**Response: The alignment process may well be useful for many readers, although the full description is probably beyond scope of this manuscript. We do have an existing 9-page document which will be available upon request for those who wish to read it. Specialist tools that would make the alignment process easier/more accurate include a large and stable optical bench with adjustable mounts, plus a more accurate laser level with a better collimated beam. The poorly collimated laser level we currently use produces a large blurred image and 2mm of movement is indeed difficult to perceive! We have rewritten the paragraph to partially explain the process and better justify our claim regarding the 0.1° accuracy figure. The new paragraph reads as follows:**

**"Without access to specialist tools, such as a large optical alignment bench and a well-collimated precision laser level, it may be difficult to accurately align mirrors in a solar tracker. For example, during our alignment process we use a bubble level, with an accuracy of about 0.02°, to initially level the tracker. We then use a self-levelling laser of similar accuracy to set a reference mirror vertical. Only then do we begin to adjust the tracker's mirrors, again using the same laser. The resulting alignment accuracy is an accumulation of the laser and bubble level uncertainties at each step in the process, plus other uncertainties involved using an oil-bath for beam reflection and the stability of ancillary optics used in the process. In-use movement and diurnal thermal cycling may**

**degrade alignment further. We estimate that overall alignment accuracy better than 0.1° is difficult to achieve or maintain in our experience.”**

*Line 157 ff: The following chapters 3.1, 3.2, and 3.3 are hard to comprehend since the information are badly structured. I'd very much appreciate an explaining sentence on how your feedback system works in the first paragraph. If I've understood it correctly, you are not actively tracking the Sun solely by the optical feedback system but are primarily relying on astronomical calculus. The optical feedback than adjusts the error offset vector in a manner that the astronomical calculations provide a sufficiently precise pointing, which is the state you call "locked". Please introduce your concept in the beginning before going into details.*

**Response: Yes, you are correct in understanding how the system works, but clearly this section describing optical feedback can be improved. Section 3 (optical feedback) has been rewritten in light of comments from both referees. Optical feedback is now explained in a clearer manner in the introductory paragraph. We have appended the revised draft for this section at the end of this document.**

*Line 177: How do you approach this trade-off in image size and intensity?*

**Response: This is indeed a trade-off. Not discussed in the draft is the also the physical size of the complete feedback optics package. A larger image will need a longer projection distance (for the same focal length lenses). In practice, the image size chosen was a function of what optics were on hand and the nature of each existing instrument installation (where there was physically room to fit the optics). The pick-off mirror can be enlarged, reducing the system noise, but at the expense of stealing some of the solar light needed for the spectrometer. Section 3 (optical feedback) has been rewritten in light of comments from both referees. We have included further discussion on design concerning image size and intensity. We have appended the revised draft for this section at the end of this document.**

*Line 183: The partly illumination of the photodiodes is important to get the basic idea of the feedback loop, mention it earlier.*

**Response: Yes indeed. Section 3 (optical feedback) has been rewritten in light of comments from both referees. The partial illumination of the diodes is now discussed earlier, and the sentence now reads:**

**"The diodes are positioned to be partially illuminated by the edge of the image, thus ensuring a high response to any movement of the image across their surface.".**

*Line 190 ff: Algorithm description hard to comprehend. Please introduce in a clear way*
*• what's the threshold and hysteresis parameter?*
*• how does the algorithm steer the motors and what happens if parameter bounds are crossed?*

**Response: Yes, the threshold and hysteresis parameters need further description and the steering process described in more detail. Section 3 (optical feedback) has been rewritten in light of comments from both referees. These threshold and hysteresis parameters are now clearly introduced early in a renamed section 3.2 "Feedback decision algorithm". We have appended the revised draft for this section at the end of this document.**

*Line 218: Name the parameters you use for the feedback correction and explain why they are sufficient.*

**Response: Addressed below**

*Line 230: Again, you use "the parameters" without naming or explaining them. This would help a lot following your characterization approach.*

**Response: Yes, we need to name the translation parameter (it was never named!) and introduce it and the rotation parameter earlier. Section 3 (optical feedback) has been rewritten in light of comments from both referees. Specifically, the translation and rotation parameters are clearly named and explained in the renamed section 3.3 "Image translation and rotation algorithm". We have appended the revised draft for this section at the end of this document.**

*Line 318 ff: Give manufacturer and description to each compartment, e.g. Bluetooth module*

**Response: The make and model of the major components are given in the figures (e.g. Fig. 7). Some parts such as the Bluetooth serial dongle and AC power supply modules are quite generic, are easily swapped with alternative but functionally equivalent versions. We currently use more than one brand amongst our existing trackers. Including such detail in the body of the manuscript risks disrupting the flow of the description. No change has been made to manuscript.**

*Line 393: On which stations are these Trackers?*

**Response: These at Lauder, New Zealand. Unfortunately, the Arrival Heights (Antarctica) spectrometer is not capable of TCCON near-IR measurements so we cannot assess tracker accuracy using the TCCON S-G shift method. It is possible to diagnose the S-G shift using MIR spectral fits, but this feature has not implemented yet. We have further identified the location of the spectrometers (and Tracker1 and Tracker 2) in the paragraph by appending:**
**"…, based at Lauder, NZ".**

*Line 404: Description of parameters must be given in algorithm chapter.*

**Response: I assume you are referring to the "offset error variables". Unfortunately, we used "error offset variable/s" 3 times in the preceding text. We have now corrected this, standardising on the "error offset" version for the manuscript.**

*Figure 13: How long are the averaging periods?*

**Response: The periods are quite variable, but we decided that a minimum of 1 hour of "likely" continuous clear sky was needed, per point, to be meaningful. On perfect blue days, in summer, this point might represent up to 12 hours of measurements. The 1-hour duration was chosen so we could plot more days (those that were largely cloud-affected) even though the data might be more variable. We have added this information to the caption by editing the line to read:**
**"Each point is a daily average value for TCCON spectral measurements taken during days that experienced at least 1 hour of continuous clear sky. "**

***Specific comments***

*Line 12/13: Use of "simple" and "just" unnecessary.*

**Response: We agree that "just" is unnecessary, and this has been removed.**
**"Simple" remains an important characteristic of this design, so it remains.**

*Line 48: Can you provide the reflectivity range of the mirrors?*

**Response: Sure – mirror reflectance (or rather the mirror coating chosen) will need to match the requirements of the spectrometer used with the tracker and the longevity required of the mirrors. We settled on using protected aluminium, which gives a good response from the visible out to about 20 microns. We added the following sentence to discuss our choice:**

**"Their reflectivity matches the typical wavelengths used for our spectrometers and the protective surface is hard-wearing, allowing for some limited cleaning.".**

*Line 59: Chapter 2 only covers mispointing of passive solar trackers, please add this.*
**Response: Thank you, we have added the word "passive" to make this clear. We have also added "…of a passive solar tracker" to the title of section 2.1.**

*Line 67: Define "vital" role, sth. like "Our trackers enable direct sun observations at sites XY..."*
**Response: We think "vital" is not needed, so it has been deleted.**

*Line 76: Define "small changes"*
**Response: We have added the line:**
**"For example, to improve our understanding of the carbon cycle we require a measurement precision of about 0.25% (e.g. Rayner and O'Brien 2001).".**

*Line 78: Add "passive" to the chapter title*
**Response: Thanks, yes clearly needed. We added "…of a passive solar tracker" to the title of section 2.1.**

*Line 97: "required" instead of "needs used"*
**Response: Thanks, change made.**

*Line 139/40: remove "simple" and replace "good" with something more descriptive*
**Response: We're reluctant to imply that no passive solar trackers could achieve better than 0.1% accuracy – and we imagine some skilled, well-resourced designers have achieved this. Hence, we use the word simple, which covers our design. We agree that "good" is better replaced by "acceptable".  This change has been made.**

*Line 158: Either remove "large" or put it into context*
**Response: We have revised the sentence to read: "…edge detection using four silicon diodes spaced evenly around the perimeter of a focused solar image of approximately 60 mm in diameter."**

*Line 169: Remove "very"*
**Response: This word was removed.**

*Line 173: "[...] adjustments performed by another mirror." The figure caption calls this the alignment mirror, since it, as explained later, is used to adjust the optical axis onto feedback plane. Be consistent with naming and explain parts at first occurrence in the text.*
**Response: This word "another" was replaced by "the alignment".**

*Line 186: What's an adequate diode size relative to image?*
**Response: The larger size solar images (~60 mm or perhaps greater in diameter) probably benefit from the active diode area being around 7 mm², which is the area of the diodes we use. The larger area gives are less noisy signal if the image is not so intense (compared to a smaller image). But also, the larger images have a less defined edge – and we feel this is better suited to broader detection surface area. Smaller images (less than 20 mm diameter) may require smaller diodes to truly edge-detect. Perhaps 1 mm² would be the smallest practical diode size although we have not**

tried this yet. Like much of the design process, there is this trade-off in total optics size, image size, pick-off mirror size, and also diode size (and placement in relation to the image edge).

with an active surface area chosen to suit the solar image size. During the rewrite of section 3 we clarified this point with the following text:

"…with an active surface area chosen to suit the solar image size. In our examples we used diodes with about 7 mm2. The diodes are positioned to be partially illuminated by the edge of the image, thus ensuring a high response to any movement of the image across their surface"

*Line 213: Sentence structure is weird, either introduce both approaches using colons or neither one, I'd go with the latter.*
**Response: We agree this is messy and have rewritten the paragraph for better clarity. We have appended the revised draft for this section at the end of this document.**

*Line 231 ff: Mention that you move the image on the feedback plane. Also, describe in further detail how you determine the angular offset.*
**Response: Section 3 (optical feedback) has been rewritten in light of comments from both referees. Specifically, we add "...at the diodes" to help identify the reference plane, and we have clarified how the rotation offset is determined.**

*Line 239: "consecutive" implies successive days, but 6 month lay in between, I'd recommend another adjective.*
**Response: "Consecutive" behaviour was a useful adjective to retain when discussing pre-correction. We have added "...as do plots six months apart" to make the reference to figure 12 more valid.**

*Line 259: Can you give a notion how reliable the software has proven to be, e.g. a number of failures during the 11 year period?*
**Response: The paragraph including this line discusses trackers in general, not our design in particular. However, your question is best answered in section 6.2 (Reliability achieved), where the value of "many months" is used. As a further guide, we don't recall ever having a software failure on one of our systems, yet another system might suffer a software crash as often as every 3 months or so. Of course, we are dealing with different computers and different versions of Linux OS so finding the exact cause is difficult. We made no changes to the text.**

*Line 261: Maybe give examples, see e.g. Heinle and Chen (2018) (doi: 10.5194/amt-11-2173-2018)*
**Response: Thank you. We have referenced this example.**

*Line 277: remove "simple"*
**Response: This word was removed.**

*Line 284: make clear what you're comparing to with "greater".*
**Response: We have edited the text to read "In addition to enhanced reliability over other methods of power transfer…".**

*Line 327: Define why the modules are "Good"*
**Response: The word is probably superfluous, as there is no reason to use poor quality (cheap) units in an instrument such as this. The words "good quality" are removed.**

*Line 390: remove "very"*

**Response: This word was removed.**

*Line 444: Your title poses a question, give a decisive answer in the following paragraph.*
**Response: The title poses a question because the answer is subjective. What period defines longevity? To help answer this we now modestly add the sentence "We believe our design demonstrates longevity.".**

*Line 467: "works very well" ! e.g. "surpasses precision requirements"*
**Response: We have altered the text to read "…feedback is effective".**

*Line 469: replace "good"*
**Response: This word was replaced with "useful".**

*Technical corrections*
*Line 55/56: space between numbers and units "2 mm", please add the space everywhere in the text.*
**Response: Thank you. All instances of this error have been corrected.**

*Line 128: horizon*t*al*
**Response: Corrected.**

*Line 287: so *IT* is ready*
**Response: Word added.**

*Line 291: present tense "use"*
**Response: Replaced by "use".**

*Line 326: "of" should be an "a"*
**Response: Changed, thanks.**

*Line 330: "[...] inside *OF* the [...]"*
**Response: Word added.**

*\*\*\*\*\*\*\*\*\*\*\*\*\*\*\*\*\*\*\*\*\*\*\*\*\*\*\*\*\*\*\*\*\*\*\*\*\*\**

[revised manuscript text omitted]

---

## Author Comment (AC2) · 3 Sep 2020

**Authors' response to Anonymous Referee #2**

We thank the reviewer for taking the time to read, comprehend and provide useful comments regarding our manuscript. In this document, the reviewer's comments are in italic, and the authors' responses follow each comment in plain bold text.

**... I share the concerns of Reviewer #1 that certain sections could be significantly improved by re-writing with a better focus on the narrative progression.**

Response: We have attempted to improve the narrative progress with re-writing and re-ordering sections of the manuscript. Specifically, section 3 (optical feedback) has been rewritten in light of comments from both referees. We have appended the revised draft for this section at the end of this document.

Also, while the authors dive in with significant details on some portions of the design, other sections read as if their inclusion was simply an afterthought (e.g., two brief sentences regarding the cover for the tracker – a critical component of any remotely operated system.)

Response: We discuss this matter in more detail below.

Lines 64-77: Section 2 Intro – I suggest improving the clarity of this section intro with a more direct connection between the question posed on line 65 (accuracy required) and the answer of 0.05 deg given on line 74.

Response: We decided to change the intro to this section, turning the question into the statement: "However, the tracker should be fit for purpose, with an accuracy that suits the intended application.".

Line 70: comma after MIR Response: Thanks, comma added

Line 98: "...needs used within..." Response: As noted by Ref #1. Replaced with "required"

Line 99: please include version of PyEphem

Response: We used version 3.7.6.0 in at least one instance. We decided not to add this detail to the text because it is not especially important.

*Line 163: "...are also simple, AND easy to implement and comprehend"* **Response: This sentence was removed during the rewrite of section 3.**

Line 201: How does the system respond to a partially obstructed solar disk (sliding along the edge of a cloud over many seconds). And how does it recover after such an episode? Could you increase the 0.001deg/sec offset adjustment based on the degree of signal difference within a sensor pair to more quickly recover?

Response: This is a good question to ask as it raises the issue of the inherent weakness of edge detection and the potential to improve the simple feedback algorithm. How the tracker responds to such a cloud episode depends on such factors as the value chosen for the "cloudiness" detection when deciding threshold parameter value, and how the diodes are placed in relation to the image edge (and their relative sizes). Without doubt, cloud obscuring any diode will affect pointing, but less so than would occur with a true quadrant diode sensor. This is because we see a sharp transition as the image edge slides off small active area the opposing diode in the pair (as compared to a larger segment area of a quadrant detector). A camera-based system should be able to perform better in this respect (compared to either edge detection or quadrant solutions).

In an FTS measurement, we really don't want any significant signal intensity change through the measurement period (which may be several minutes). Our preference is to have the threshold parameter set so that the measurements get flagged as "cloudy" in the daily log file (and the measurements are later removed). In post processing we further filter on detecting an intensity change. In either case we remove the measurement, which to a large degree, removes measurements with cloud induced mispointing.

There is much potential to improve the algorithm. We have trialled changing the step-size based on sensor-pair error difference, but it meant some customising was necessary for each of our installations (they are all very different – something we aim to change). A better approach may be to use a form of PID control loop, or switch to PID once initial lock has been established. This is something we are very keen to try. In the meantime, the simple system we use works well enough.

We have broadened the discussion on these potential improvements within section 7.1, by specifically mentioning PID control and dynamic step size:

"This could also be sped-up by saving and reusing offsets from a recent clear day, or dynamically changing step size depending on the magnitude of the diode pair difference."

**We also added a discussion about cloud:**

"An inherent weakness with the edge detection method is the response to a partial obscured solar disk. Under this situation the pointing is affected to some degree, but less so than would occur with a quadrant detector. A camera should be able to behave better in this situation.".

**Line 230: Reference to webserver before the webserver is introduced.**

Response: Thanks. This is corrected in the rewrite for section 3: "This can be done using the application's buttons on-screen (see Sect. 5.1 which discusses the webserver, and also Fig 6.).".

**Line 250: The threshold level setting procedure / choice of hysteresis parameter is not particularly clear. How do you define "works well" when set at 10%?**

Response: We clarify the setting of this parameter in the rewrite of section 3, for expample: " The hysteresis parameter works well if set at about 10 - 30% of the threshold value and seems to have little effect on accuracy if the image perimeter and diode positions are well matched.".

**Line 260: drop s from mains?**

Response: We have replaced the word "mains" with the more universal "AC".

Line 288: Does simply passing the reference point in normal operating mode sufficiently maintain the rotator stage's knowledge of its position? While such knowledge may not be necessary in active tracking, is it not still needed in the passive ephemeris tracking mode?

Response: We think you are asking "is the ephemeris used during passive mode (and thus at night?)" If so, the answer is yes. The software also remembers the rotator position at all times. At sunrise, active tracking will then again start fine-tuning the pointing based on adding an offset to the rotator position.

Section 3 (optical feedback) has been rewritten in light of comments from both referees. Specifically, we have clarified the operation of passive and optical feedback modes in the revised text. We have appended the revised draft for this section at the end of this document.

**Line 333: Space after Fig.* **Response: Added thanks.**

**Line 367: Why is the tracking accuracy not regularly analysed for the other two trackers you have built?**

Response: The other three trackers are used with Mid-IR spectrometers which don't normally measure in the wavelengths we use to retrieve TCCON S-G shift. Two of the spectrometers can have beam splitters and detectors exchanged so the required Near-IR measurements can be made but this is invasive, and only done occasionally (e.g. for the spectrometer intercomparison in 2019). We have changed the sentence to correctly read:

"Tracker accuracy has been analysed on our trackers capable of TCCON measurements and can routinely produce a pointing accuracy of 0.02° from solar centre.".

The other two trackers haven't had analysis using S-G shift and are not discussed individually in terms of accuracy, although we expect they perform well judging by image monitoring.

Line 370: While I applaud the addition of the wall image in the design – certainly a very useful visual for a lab operator in the room – I would hardly refer to it as a valid means of monitoring the tracking accuracy, especially for a design which you emphasize can be efficiently operated remotely. I recommend de-emphasizing the quick visual method, and perhaps more clearly highlighting the consistency of the offset errors as a function of alt/az.

Response: The are a few points raised in this comment. Firstly, we cannot take credit for the wall image idea and it's not part of the tracker design at all. However, it is quick and easy method of checking that the tracker is functioning (and accurate) and that the sun is clear. We use it daily. The eye and brain can tell, in a glance, that all is well with the tracker. The next best method to quickly assess accuracy is to view the spectrometer input aperture (if you can see it, which would need a transparent cover added). But this assumes that the 45-edgree mirror (which not strictly part of the tracker) is adjusted correctly to centre the image on the aperture. Indeed, this operator error is responsible for many of the large jumps in S-G shift plotted in Figure 13. This is discussed in line 505 of the draft. The tracker itself probably performs with a higher consistent accuracy than is indicated by Figure 13 – but until we physically attach the feedback to the spectrometer, we can't see this by using the S-G shift method.

After producing the Figure 13, and seeing these jumps, we decided to advance our work schedule and have now achieved a working prototype feedback optic attached to the instrument. We have trialled this by intentionally moving the instrument and watching as the feedback steers the image back to the centre of the aperture. This will undoubtably improve long-term retrieved pointing accuracy and spectrometer data quality, although in itself may not improve the accuracy of the tracker (when considered as a unit on its own).

In the draft, we refer to "remote" twice when discussing reliability and once when discussing access control over the network using the webserver interface. To assess accuracy in a truly remote site, with no staff present, the best method remains S-G shift (if these measurements are possible), or via a camera constantly viewing of the input aperture. However, with no humans in the remote lab to cause problems, we are quite confident that edge detection method will remain accurate no matter whether the feedback optics are on the tower or the spectrometer. Tracker performance can then be monitored periodically by looking at offsets, the diode solar intensities (logged each second) or the webserver.

The alt/az offset errors remain a useful diagnostic as you say and are readily available at any time. With some extra effort, it might be possible to extract the nature of the error (alignment or levelling). If so, this would be even better.

**We have considered this comment carefully but decided to leave the text unchanged.**

**Line 410: Given that the offset errors are a function of alt/az, how useful is a single alt/az setup offset parameter pair? I imagine these much be updated throughout the year?**

Response: It's true these errors offsets are some function of alt/az. But they also have an underlying fixed offset component which comes from the "random" uncertainty when the tracker was initially parked during the last powered-up (the parking error resulting from a single coarse flag). It makes sense to zero this at some stage, but inevitably it will be done at different alt/az than last time (possibly months ago). Hence the offset of the plots clearly seen in Figure 12. Comparing plots of errors offsets with radically different day lengths would also appear a bit awkward, but (by pure good fortune) the months plotted in figure 12 have similar solar characteristics. We added: "...similar day length..." to the sentence discussing figure 12.

In terms of using error offset values to "pre-correct" (line 240 in draft) then yes, the offsets would lose their value after a few weeks. The pre-correction would need to either be a dynamic process (using values from recent clear days) or use a historical (annual) dataset or fitted function. Both are valid options worth trying, especially if we needed more precise passive tracking (e.g. lunar observations).

**Line 516: This manuscript regularly emphasizes the remote operation of this tracker. Thus, a reliable automated cover is a critical component of the design. I believe it would be very appropriate and desirable to include more information on the cover design in this manuscript.**

Response: We agree that the cover is a critical component of the complete observation system and another potential weakness that needs careful design and construction. Typically, a cover is designed to "match" the tracker, which in this case now rotates 360 degrees, continuously! While we would love to include a full description of this design (successfully in use with two of our four trackers), we feel it is beyond scope for this manuscript. However, we do feel encouraged to consider it a candidate for future publication, perhaps as a technical note, in the near future. In terms of a cover for a remote site, then the reliability of the design takes on even more importance. We have added the line:

"The authors would like to present the design of this cover in a future publication."

**Fig 13: Despite being well within the required accuracy requirements, I would like the authors to speak to the changes in S-G parameter seen in November 2018 and in early Jan 2019 ahead of the optics cleaning**

Response: We agree these are major steps in retrieved S-G shift. Unfortunately, the exact cause cannot be determined because not every disturbance in the laboratory is logged. For example, we often have tour groups of visitors through the lab, or other staff. The lab space is very tight. Often someone will lean on or bump the large spectrometer instruments (or worse, bump the tower 45-degree mirror). We expect these disturbances to be less of a problem now that the feedback optics are physically attached to the spectrometer. We have added the following sentence to the section discussing figure 13:

[revised manuscript text omitted]